



# Relationships between leaf δ[15]N and leaf metallic nutrients
Chongjuan Chen[1,2]   Yingjie Wu[1]   Shuhan Wang[3]   Zhaotong Liu[1]   Guoan Wang[1]*
[1]Beijing Key Laboratory of Farmland Soil Pollution Prevention and Remediation, Department of
Environmental Sciences and Engineering, College of Resources and Environmental Sciences, China
Agricultural University, Beijing, 100193, China.
[2]Institute of Surface-Earth System Science, Tianjin University, Tianjin, 300072, China
[3]Institute of Vegetables and Flowers, Chinese Academy of Agricultural Sciences, Beijing, 100081,
China
Correspondence to: Guoan Wang (gawang@cau.edu.cn)
## Abstract
Metallic nutrients play a vital role in plant physiological and biochemical processes such as nitrogen
uptake and assimilation, which cause isotopic fractionation against [15]N. Thus, investigating the
relationships between leaf nitrogen isotope ratio (δ[15]N) and leaf metallic nutrients could enhance our
understanding of nitrogen (N) cycling. However, to our knowledge, these relationships have not been
examined as yet. To fill this research gap, we analyzed leaf δ[15]N and leaf potassium (K), Calcium (Ca),
magnesium (Mg), iron (Fe), manganese (Mn) and zinc (Zn) contents of 624 non-$N_2$-fixing plant
samples and revealed the relationships between leaf δ[15]N and these metallic elements. Overall, leaf δ[15]N
was positively correlated with leaf K, Ca, Mg and Zn, negatively correlated with leaf Fe, and not related
to leaf Mn. The relationships between leaf δ[15]N and leaf K, Ca, Mg, Zn and Fe were not affected by
both vegetation type and soil type, suggesting that the observed relationships could be universally valid.





However, the relationship between leaf $\delta^{15}N$ and leaf Mn depended on vegetation type and soil type,
therefore, the observed relationship should not be considered to be universal. These metallic nutrients
together accounted for 55.7% of the variations in leaf $\delta^{15}N$; this emphasized the significance of metallic
nutrients in determining leaf $\delta^{15}N$. To the best of our knowledge, this is the first study which adressed
the relationships between leaf $\delta^{15}N$ and leaf meteallic nutrients. However, further investigations are
needed to reveal the underlying mechanism.

## 28 1 Introduction

Nitrogen (N) cycling has received considerable attentions because that N is deemed as the key element
in regulating productivity of terrestrial ecosystems (Fay et al., 2015; Wieder et al., 2015) and many
nitrogenous compounds, such as $N_2O$, NO, or $NH_3$, generating from N cycling link to a series of major
environment issues (Bourgeois et al., 2018; Desmit et al., 2018). Nitrogen isotopic composition in leaf
(leaf $\delta^{15}N$) was usually regarded as an integrator of terrestrial N cycling (Houlton et al., 2006, 2007;
McLauchlan et al., 2007, 2013; Robinson, 2001). Revealing the potentially influential factors of leaf
$\delta^{15}N$ and investigating the relationships between leaf $\delta^{15}N$ and these factors could help to strengthen our
understanding of N cycling (Craine et al., 2009; Hobbie and Högberg, 2012).
Many attentions have been paid to the variations in leaf $\delta^{15}N$ associating with precipitation (Handley
et al., 1999; Robinson, 2001; Amundson et al., 2003; Craine et al., 2009), temperature (Martinelli et al.,
1999; Amundson et al., 2003; Craine et al., 2009; Sheng et al., 2014; Yang et al., 2013), soil N
availability (Houlton et al., 2006, 2007), altitude (Liu et al., 2010; Liu and Wang, 2010) and mycorrhiza
association (Hobbie and Colpaert, 2003; Hobbie et al., 2008; Hobbie and Högberg, 2012). In addition to





these influential factors, mineral nutrients play an essential role in N cycling as well (Marschner, 2012).
Although previous studies have exposed the relationships between leaf $\delta^{15}N$ and leaf mineral nutrients
(Craine et al., 2005, 2009; Pardo et al., 2006), all of these studies have simply investigated the
correlations between leaf $\delta^{15}N$ and leaf nitrogen (N) and phosphorus (P) because N and P play a vital
role in plant growth (Han et al., 2011; Vitousek et al., 2010). To our knowledge, there is no report
examining the relationships between leaf $\delta^{15}N$ and leaf metallic nutrients, such as potassium (K),
Calcium (Ca), magnesium (Mg), iron (Fe), manganese (Mn) and zinc (Zn). The demands of metallic
nutrients are lower relative to N and P for plant growth, but metallic nutrients also play a fundamental
role in plant physiological function and biological chemistry (Marschner, 2012; Vitousek et al., 2010)
and are involved in N cycling (Armengaud et al., 2009; Epstein and Bloom, 2005; Marschner, 2012).
K is the activator of many enzymes in plants; it promotes photosynthesis and absorption of N of
plants, especially for nitrate utilization (Coskun et al., 2017; Zhang et al., 2010). Ca is a ubiquitous
secondary messenger involved in many physiological processes and also plays a role in nitrate signaling
(Krouk et al., 2017; Liu et al., 2017). Mg ion plays an vital function in nitrate reduction and affects
synthesis of protein, and it is an essential component of chlorophyll as well (Bose et al., 2011). Fe
participates in many physiological processes in plants, such as nitrogen assimilation, photosynthesis,
respiration, DNA synthesis, hormone and coenzyme synthesis (Balk and Pilon, 2011; Shokrollahi et al.,
2018). Mn is the cofactor and activator of some key enzymes in plants, including malic dehydrogenase,
DNA synthetase, RNA synthetase and nitrite reductase (Mukhopadhyay and Sharma, 1991). Zn is
involved in protein synthesis, auxin metabolism and carbohydrate metabolism, and so on (Henriques et
al., 2012). Overall, these metallic nutrients are involved in plant N uptake and assimilation. Since





nitrogen isotopic fractionation is associated with the process of plant N uptake and assimilation (Evans,
2001; Tcherkez and Hodges, 2008; Liu et al., 2014), we hypothesize that leaf $\delta^{15}N$ relates to these leaf
metallic nutrients. Thus, the aim of the current study was to confirm the hypothesis by measuring leaf
$\delta^{15}N$ and leaf K, Ca, Mg, Fe, Mn and Zn contents of more than 600 plant samples from mainland China.

## 68  2 Materials and methods

### 69  2.1 Study area

This study was conducted along the 400 mm isohyet in China which extends from the southern slope of
Greater Khingan in northeast China, passing through the Tai-hang Mountains, to the eastern part of
Qinghai-Tibet Plateau in southwest China. This study used the same sampling transect and sites as Tan
et al. (2019). Fifty-eight sampling locations were set along the 400 mm isohyet from Luoguhecun (site
No. 1, 53.29 °N, 122.15 °E) of Heilongjiang Province in northeast China to Zhanang (site No. 58,
31.41 °N, 91.96 °E) of Tibet in southwest China (Fig. S1, Table S1). Among these locations, the lowest
mean annual temperature is -5.1 ℃ (site No. 55, Qumalai) and the highest is 9.7 ℃ (site No. 41,
Hengshan-1) (Table S1). The average mean annual precipitation of these sampling locations is 397.2
mm (Table S1).

### 80  2.2 Plant and soil sampling

Plant leaves were sampled in the summer of 2008 and 2013. Plant samples were collected at locations
that are far away from human habitat and major roads to minimize the effects of shading and human
activities. Almost all plant species at each location were collected. For each plant species, the same



number of leaves were collected from 5 – 7 individual plants; plant leaves of the same species from
each site were combined into one sample. For shrub and herb species, the uppermost leaves were
sampled; for tree species, 2 leaves at each of the 4 cardinal directions about 8 – 10 m above the ground
were collected. A total of 658 plant samples were collected along the 400 mm isohyet, including 624
non-$N_2$-fixing plant samples and 34 $N_2$-fixing plant samples.

**2.3 Laboratory measurements**
The clean and dried plant sample was ground into a fine powder using a planetary mill with a 40 mesh
screen. 1.700 mg to 2.000 mg of plant sample  was weighed in tin capsules using an electric balance
with a precision of $10^{-6}$ g (ME 5,Sartorius Genius Series, German). $\delta^{15}N$ and N contents in leaves were
determined by a Delta$^{Plus}$ XP mass spectrometer (Thermo Scientific, Bremen, Germany) coupled with
an automated elemental analyzer (Flash EA1112, CE Instruments, Wigan, UK) in a continuous flow
mode at the Stable Isotope Laboratory of the College of Resources and Environmental Sciences, China
Agricultural University. The nitrogen isotope values were expressed in the standard notation relative to
air $N_2$ using the equation:
$$\delta^{15}N\ (‰) = (^{15}N/^{14}N_{sample} / \ ^{15}N/^{14}N_{air} - 1) \times 1000$$
the standard deviations of N contents and $\delta^{15}N$ were less than 0.1% and 0.15‰, respectively, among
replicate measurements of the same sample.
The powdered leaf samples were accurately weighed (0.2000 g) and placed into the bottom of the
microwave digestion tube, 6 mL concentrated nitric acid was added into the tube, the tube was shaken
well and settled for overnight. The next day 2 mL $H_2O_2$ was added into the tube and the mixture was





blended well. The mixture was digested to clear solution with no obvious residue by a microwave
digestion oven (MARS Xpress, CEM, USA). The digested solution was completely moved to a
volumetric flask and diluted to 25 mL. After the solution was shaken well and settled for 30 minutes,
the clear solution in the upper was moved to a 10 mL centrifuge tube and saved to determine its contents
of mineral elements. The standard substance for this measurement was Henan wheat (CAS number:
GSB-24). The measurement procedure of the standard matters was the same as that of sample. Two
samples of the standard matters were used for every forty samples measured. Meanwhile, three blank
control groups were carried out in the same process. The standard deviations of contents of mineral
elements were less than 5%. The contents of leaf K, Ca, Mg, Fe, Mn and Zn were measured by
ICP-OES (7300 DV, PerkinElmer, USA) at wavelengths of 766.5 nm, 317.9 nm, 285.2 nm, 238.2 nm,
257.6 nm, 206.2 nm, respectively.

**2.4 Statistical analysis**


The contents of leaf metallic nutrients were log-transformed to improve data normality. As two axes
were equally prone to error, to avoid biases of the slope estimates, reduced major axis (RMA)
regression was applied to detect the linear relationships between leaf $\delta^{15}N$ and leaf metallic nutrients
and the relationships between leaf N and leaf metallic nutrients. Partial correlation analyses were
conducted to evaluate effects of soil type and vegetation type on the relationships between leaf $\delta^{15}N$ and
leaf metallic nutrients. Multiple linear regressions were used to detect the influences of leaf metallic
nutrients on leaf $\delta^{15}N$. All statistical analyses were conducted by SPSS software (SPSS for Windows,
Version 20.0, Chicago, IL, USA) with a significance level of $P < 0.05$.




## 3 Results

### 3.1 Relationships between leaf $\delta^{15}N$ and leaf metallic nutrients for all non-$N_2$-fixing plant species

The mean contents of leaf metallic nutrients in all plant species (including $N_2$-fixing and non-$N_2$-fixing plants) were reported by Tan et al. (2019) in which the same set of plant samples were used. The contents of leaf K, Ca, Mg, Fe, Mn and Zn for all non-$N_2$-fixing plants ranged from 451 to 111874 mg/kg, 874 to 67980 mg/kg, 310 to 34673 mg/kg, 35 to 9181 mg/kg, 10 to 2476 mg/kg and 5 to 226 mg/kg, with an average of 22365 mg/kg, 18114 mg/kg, 5656 mg/kg, 764 mg/kg, 115 mg/kg and 45 mg/kg, respectively. Reduced major axis (RMA) regression showed that leaf $\delta^{15}N$ increased with leaf K, Ca, Mg and Zn ($P < 0.001$ for leaf K; $P < 0.001$ for leaf Ca; $P < 0.001$ for leaf Mg; $P < 0.001$ for leaf Zn) and decreased with leaf Fe ($P < 0.001$), whereas leaf $\delta^{15}N$ did not exhibit obvious change trend with leaf Mn ($P > 0.05$) (Fig. 1). Multiple linear regression suggested that 54.3% of the variability in leaf $\delta^{15}N$ could be explained by the combination of leaf K, Ca and Mg (see model-1 in Table 1), and 15.0% by the combination of leaf Fe, Mn and Zn (see model-2 in Table 1), and 55.7% by the combination of all these six metallic elements (see model-3 in Table 1).

Since the sampling spanned a vast geographic scale and involved a variety of vegetation types and soil types, the observed relationships between leaf $\delta^{15}N$ and leaf metallic nutrients could be affected by vegetation types and soil types. To determine whether the two factors exerted an influence on the relationship, we conducted a series of partial correlation analyses in which vegetation type and/or soil type were controlled. The partial correlation analyses of leaf K, Ca, Mg, Fe and Zn vs. $\delta^{15}N$ yielded almost the same results as bivariate correlation analyses did, whereas significant changes observed in





the relationship between leaf $\delta^{15}N$ and leaf Mn, i.e. the relationship was not significant in bivariate
correlation analysis, but it became significant in the partial correlation analyses after soil type was
controlled (Table 2).
RMA regression analyses showed that leaf N was positively correlated with leaf K, Ca, Mg and Zn
(all $P < 0.001$), negatively with leaf Fe ($P < 0.01$), whereas not with leaf Mn ($P > 0.05$) (Fig. 2). The
relationships between leaf N and leaf metallic nutrients were similar to the relationships between leaf
$\delta^{15}N$ and leaf metallic nutrients (Fig. 1 and 2). Partial correlation analyses were conducted to examine
the relationships between leaf $\delta^{15}N$ and leaf metallic nutrients after controlling for leaf N. Compared
with the relationships between leaf $\delta^{15}N$ and leaf K, Mg, Fe, Mn and Zn without controlling leaf N, the
relationships between them were still significant although the relationships became weak after leaf N
was controlled (Table S3). However, the relationships between leaf $\delta^{15}N$ and leaf Ca vanished after leaf
N was controlled (Table S3).

**3.2 Relationships between leaf $\delta^{15}N$ and leaf metallic nutrients at plant functional group level**

The relationships between leaf $\delta^{15}N$ and leaf metallic nutrients at plant functional group level were
examined, considering considerable differences of leaf $\delta^{15}N$ and leaf metallic nutrients across plant
functional groups (Chen et al., 2017; Han et al., 2011; Tan et al., 2019). Leaf $\delta^{15}N$ in herbs, annual herbs
and perennial herbs positively correlates to leaf K (all $P < 0.001$), whereas leaf $\delta^{15}N$ in woody plants did
not relate to leaf K ($P > 0.05$) (Fig. 3). Leaf $\delta^{15}N$ in herbs and annual herbs increased with leaf Ca ($P <$
0.001 for herbs and $P < 0.01$ for annual herbs), whereas leaf $\delta^{15}N$ in perennial herbs and woody plants
did not change with leaf Ca (both $P > 0.05$) (Fig. 3). Leaf $\delta^{15}N$ was positively related with leaf Mg for



all plant functional groups ($P < 0.001$ for both herbs and annual herbs, $P < 0.01$ for both perennial herbs
and woody plants) (Fig. 3). Leaf $\delta^{15}N$ increased with leaf Fe in herbs, annual herbs and perennial herbs
($P < 0.001$ for both herbs and annual herbs, $P < 0.05$ for perennial herbs), whereas kept constant with
leaf Fe in woody plants ($P > 0.05$) (Fig. 3). Leaf $\delta^{15}N$ correlated positively and negatively to leaf Mn in
annual herbs and perennial herbs, respectively (both $P < 0.05$), while did not correlate with leaf Mn in
herbs and woody plants (both $P > 0.05$) (Fig. 3). Leaf $\delta^{15}N$ was positively related to leaf Zn in herbs and
annual herbs (both $P < 0.001$), whereas was not related to leaf Zn in perennial herbs and woody plants
(both $P > 0.05$) (Fig. 3).

**177   3.3 Relationships between leaf $\delta^{15}N$ and leaf metallic nutrients in widely distributed genera and**

**178   species**

To further address the relationships between leaf $\delta^{15}N$ and leaf metallic nutrients, we also did the same
investigation on two common genera (*Artemisia* and *Chenopodium*) that widely distributed in the
sampling regions. Leaf $\delta^{15}N$ in both *Artemisia* and *Chenopodium* positively correlated to leaf K (both $P$
$< 0.001$) (Fig. 4). Leaf $\delta^{15}N$ in *Artemisia* decreased with leaf Ca ($P < 0.01$), whereas leaf $\delta^{15}N$ in
*Chenopodium* did not vary with leaf Ca ($P > 0.05$) (Fig. 4). Leaf $\delta^{15}N$ was not related with leaf Mg in
*Artemisia* ($P > 0.05$) but positively with leaf Mg in *Chenopodium* ($P < 0.001$) (Fig. 4). Leaf $\delta^{15}N$
maintained constant with leaf Fe in *Artemisia* ($P > 0.05$), whereas decreased with leaf Fe in
*Chenopodium* ($P < 0.05$) (Fig. 4). Leaf $\delta^{15}N$ kept unchanged with leaf Mn and Zn in both *Artemisia* and
*Chenopodium* (both $P > 0.05$) (Fig. 4).

The variations in leaf $\delta^{15}N$ with leaf metallic nutrients were also examined in three most widespread





plant species (*Amaranthus retroflexus*, *Plantago depressa* and *Setaria viridis*) grown in the study
regions. Leaf $\delta^{15}$N in both *Amaranthus retroflexus* and *Setaria viridis* increased with leaf K ($P < 0.05$
for *Amaranthus retroflexus*, $P < 0.001$ for *Setaria viridis*), whereas leaf $\delta^{15}$N in *Plantago depressa* kept
constant with leaf K ($P > 0.05$) (Fig. 5). Leaf $\delta^{15}$N in both *Amaranthus retroflexus* and *Setaria viridis*
did not vary with leaf Ca (both $P > 0.05$), but leaf $\delta^{15}$N in *Plantago depressa* decreased with leaf Ca ($P$
$< 0.05$) (Fig. 5). Leaf $\delta^{15}$N was not related to leaf Mg in both *Amaranthus retroflexus* and *Setaria viridis*
($P > 0.05$) but positively related to leaf Mg in *Plantago depressa* ($P < 0.01$) (Fig. 5). Leaf $\delta^{15}$N was
invariant with leaf Fe and Mn for the three species (all $P > 0.05$) (Fig. 5). Leaf $\delta^{15}$N in *Amaranthus*
*retroflexus* positively correlated to leaf Zn ($P < 0.01$), whereas leaf $\delta^{15}$N in both *Plantago depressa* and
*Setaria viridis* showed no trends with leaf Zn (both $P > 0.05$) (Fig. 5).

## 4 Discussion

N acquisition of N$_2$-fixing plants was independent of soil N dynamics and $\delta^{15}$N values of N$_2$-fixing
plants were often reported to be close to 0 ‰ (Högberg, 1997; Craine et al., 2009); in addition, the
number of N$_2$-fixing plant samples was small in this study. Thus, we only addressed the relationships
between leaf $\delta^{15}$N and leaf metallic nutrients for non-N$_2$-fixing plant species. For all non-N$_2$-fixing plant
species pooled together, this study observed significant and clear correlations between leaf $\delta^{15}$N and leaf
metallic nutrients except leaf Mn, in which leaf $\delta^{15}$N showed increasing trends with leaf K, Ca, Mg and
Zn, and a decreasing trend with leaf Fe (Fig. 1). To our knowledge, this is the first exploration of the
relationships between leaf $\delta^{15}$N and leaf metallic nutrients.
The relationships between leaf $\delta^{15}$N and leaf metallic nutrients, except leaf $\delta^{15}$N and leaf Mn, almost





keep unchanged whether vegetation type and soil type were controlled or not (Table 2). This suggested
that the patterns of the variations in leaf $\delta^{15}$N with leaf K, Ca, Mg, Fe and Zn were independent of
vegetation type and soil type. However, the relationships between leaf $\delta^{15}$N and leaf Mn were associated
with soil type based on partial correlation analysis (Table 2). Therefore, the present study demonstrated
that the positive relationships between leaf $\delta^{15}$N and leaf K, Ca, Mg and Zn and the negative
relationship between leaf $\delta^{15}$N and leaf Fe may be universal; but the relationship between leaf $\delta^{15}$N and
leaf Mn will be different across sites or regions.
The variations in leaf $\delta^{15}$N with leaf metallic nutrients in herbs were similar to the results derived
from the whole non-$N_2$-fixing plant samples (Fig. 1 and Fig. 3). However, except leaf Mg and Mn,
woody plants showed different patterns of the variations in leaf $\delta^{15}$N with leaf K, Ca, Fe and Zn from
that of the whole plant sample (Fig. 1 and Fig. 3). This could be associated with the different nutrient
recycling processes or resorption abilities between herbs and woody plants (Vergutz et al., 2012), which
would influence the level of leaf metallic nutrients and cause different N isotopic fractionation between
herbs and woody plants. In herbs and many perennial herbs, nutrients are absorbed and assimilated in a
straightforward manner since little plant biomass turnover is observed. Whereas, in woody species,
considerable nutrient recycling is observed for mobile elements, including N, K, P, etc. Therefore, herbs
and woody plants would have different patterns of variations in leaf $\delta^{15}$N with leaf metallic nutrients.
The observed changes in leaf $\delta^{15}$N with leaf metallic nutrients of the whole non-$N_2$-fixing plant samples
mainly represent the variations of herb samples because most samples were herbs; it was reasonable that
woody plants showed different patterns from those for the whole samples. However, the sample size of
the woody plants was limited (47 woody plant samples), we can not give a definite conclusion about





whether the variation trend of leaf $\delta^{15}N$ with leaf metallic nutrients is dependent of plant functional
group or not.
The relationships between leaf $\delta^{15}N$ and leaf metallic nutrients were also addressed in two common
genera and three widespread species. In general, only the relationships between leaf $\delta^{15}N$ and leaf K and
Mn derived from whole non-$N_2$-fixing plant sample could be maintained at genus and species level (Fig.
4 and Fig. 5). This may be attributed to the differences in nutrient absorption capacity and nitrogen
isotope fractionation between plants. Thus, a lot of investigations on widespread genera and species will
be needed to reveal the variations in leaf $\delta^{15}N$ with leaf metallic nutrients at genera and species levels.
54% and 15% of the variances in leaf $\delta^{15}N$ were explained by the combinations of leaf K, Ca, Mg
(model-1) and Fe, Mn, Zn, respectively (model-2) (Table 1). This suggested that leaf K, Ca, Mg were
the major factors driving $\delta^{15}N$ changes. Leaf Fe, Mn and Zn also exerted significant influences on the
variations of leaf $\delta^{15}N$ even though their contributions to leaf $\delta^{15}N$ were less than that of K, Ca and Mg
(Table 1). The contribution of leaf metallic nutrients to leaf $\delta^{15}N$ depended primarily on their relevance
degree to leaf $\delta^{15}N$ (Fig. 1). Compared to model-1, model-3's interpretation of leaf $\delta^{15}N$ only increased
by 1.4% (Table 1). The reason was that the contributions of Fe, Mn, and Zn to leaf $\delta^{15}N$ were partly
included in the contributions of K, Ca and Mg, because there were strong relationships among the six
elements (Table S2).
Metallic nutrients could exert influences on the variations of leaf $\delta^{15}N$ via regulating plant N
utilization. The regulation might be associated with the role of metallic nutrients in the utilization of
nitrate ($NO_3^-$) and ammonium ($NH_4^+$) (Coskun et al., 2017; Fan et al., 2017; Maathuis, 2009). Piao et al.
(2017) reported a positive relationship between leaf K and $\delta^{15}N_{leaf-soil}$ (leaf $\delta^{15}N$ - soil $\delta^{15}N$). A similar





result was also found in the present study (Fig. S1). Furthermore, leaf $\delta^{15}N$ was correlated strongly and
positively with $\delta^{15}N_{leaf-soil}$ (Fig. S2). This resulted in the observed positive relationship between leaf
$\delta^{15}N$ and leaf K in this study (Fig. 1). K not only participated in $NO_3^-$ uptake and translocation (Coskun
et al., 2017), but also as an activator for nitrate reductase and was required for the synthesis of nitrate
reductase (Armengaud et al., 2009). $NO_3^-$ utilization could be promoted by $K^+$ (Britto and Kronzucker,
2008; Coskun et al., 2010, 2017), and the tight association between $K^+$ contents and $NO_3^-$ uptake and
transport in plants was observed in many reports (Triplett et al, 1980; Zhang et al., 2010; Drechsler et al.,
2015). Conversely, the uptake and translocation of $NH_4^+$ were inhibited by $K^+$ due to the same charge
(Touraine et al., 1988). Additionally, the relative dependence of plants on soil $NO_3^-$ or $NH_4^+$ could cause
the fractionations between leaf and roots, then change leaf $\delta^{15}N$ (Bustamante et al., 2004). Usually, there
were nearly no fractionations between leaf and roots when $NH_4^+$ is the sole source, because $NH_4^+$ is
easily and totally assimilated in roots (Piao et al., 2012; Raven et al., 1992). However, $NO_3^-$ could be
assimilated both in roots and leaf, and the unassimilated and $^{15}N$-enriched $NO_3^-$ in roots would be
translocated to leaf, so leaf $\delta^{15}N$ is higher than root $\delta^{15}N$ when $NO_3^-$ is the sole source (Yoneyama and
Kaneko, 1989; Evans et al., 1996; Kolb and Evans, 2002). So, leaf might become $^{15}N$-enriched
gradually with the increase in the dependence on $NO_3^-$ in plants. The positive relationships between leaf
K and $\delta^{15}N_{leaf-soil}$ accompanied with the positive relationship between leaf N and leaf K suggested that
plants may have a preference on $NO_3^-$ relative to $NH_4^+$ (Piao et al., 2017). Thus, the observed
relationship between leaf $\delta^{15}N$ and leaf K might also be attributed to plant's preference for $NO_3^-$.
The positive relationships between leaf $\delta^{15}N$ and leaf Ca and Mg might be associated with
substitution of $Ca^{2+}$ and $Mg^{2+}$ for $K^+$ in the charge balancing of $NO_3^-$ (Förster and Jeschke, 1993;



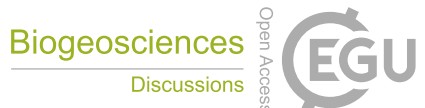

Drechsler et al., 2015). Roosta and Schjoerring (2007) demonstrated that plants would accumulate more
Ca and Mg nutrients with higher $NO_3^-/NH_4^+$ ratio in growth media. The relationship between leaf $\delta^{15}N$
and leaf Zn might be due to the function of Zn in some enzymes which participate in protein
metabolism (Henriques et al., 2012). As the crucial components, Fe and Mn constitute the structures of
N assimilatory enzymes (Fischer et al., 2005; Ventura et al., 2013). However, at present, how to explain
these observed relationships between leaf $\delta^{15}N$ and leaf Zn, Fe and Mn seems to be more challenging.
Leaf N was found to be correlated with both leaf $\delta^{15}N$ and metallic nutrients except Mn (Fig. 2), thus,
the correlations between leaf $\delta^{15}N$ and metallic nutrients were expected to be related to plant nutrient
status. However, controlling for leaf N did not fundamentally change the correlations between leaf $\delta^{15}N$
and metallic nutrients except the relationships between leaf $\delta^{15}N$ and leaf Ca (Table S3), suggesting that
the correlations between leaf $\delta^{15}N$ and metallic nutrients were usually not dependent on plant N status.
Many soil factors, such as soil organic matter, soil pH, soil C/N, soil density and so on could also
affect the variations of leaf $\delta^{15}N$ (Criane et al., 2009; Pardo et al., 2006; Robinson, 2001) and leaf
metallic nutrients (Bartuska and Ungar, 1980; Sahrawat, 2016). However, in this study, except that leaf
Mn and Zn correlated obviously to those soil factors, almost no significant relationship between those
soil factors and leaf $\delta^{15}N$ and leaf metallic nutrients was found (Table S4). Soil $\delta^{15}N$ was correlated with
both leaf $\delta^{15}N$ and leaf metallic nutrients (Table S5), thus, soil $\delta^{15}N$ might be a driver for the
relationships between leaf metallic nutrients and leaf $\delta^{15}N$. Whereas, the relationships between leaf $\delta^{15}N$
and leaf metallic nutrients were almost not changed after controlling for soil $\delta^{15}N$ (Table S6), this
suggested that the relationships between them were not related to soil $\delta^{15}N$.





## 5 Conclusion

This study revealed the relationships between leaf $\delta^{15}N$ and leaf K, Ca, Mg, Fe, Mn and Zn by investigating 624 non-$N_2$-fixing plant samples in China. Leaf $\delta^{15}N$ was positively related to leaf K, Ca, Mg and Zn, and negatively related to leaf Fe, whereas was not related to leaf Mn. Together, these leaf metallic nutrients could account for 55.7% of the variations in leaf $\delta^{15}N$, which demonstrated the fundamental role of leaf metallic nutrients in leaf $\delta^{15}N$. The relationships between leaf $\delta^{15}N$ and leaf K, Ca, Mg, Fe and Zn were independent of vegetation type and soil type, suggesting that the observed relationships could be universal. However, the relationship between leaf $\delta^{15}N$ and leaf Mn depended on soil type, which indicated that the relationship was not a general pattern. The relationships between leaf $\delta^{15}N$ and leaf metallic nutrients were not changed considerably when leaf N or soil $\delta^{15}N$ was controlled, this might indicate that these observed relationships were not dependent of plant N status and soil $\delta^{15}N$.

**Data availability**. There is no underlying material and related items in this paper. The data will be provided online.

**Author Contributions.** CC and GW designed the study. GW and SW collected the samples. CC, YW and ZL measured the data. CC and GW wrote the paper.

**Competing financial interests**. The authors declare no competing financial interests.

**Acknowledgments.** This study was supported by the National Natural Science Foundation of China





(No. 41772171). We would like to express our deep thanks to Yan Ma for determining nitrogen isotopic composition of leaves and soils at the Stable Isotope Laboratory of the College of Resources and Environment, China Agricultural University. We appreciated Dr. Muhammed Adeel in China Agricultural University for helping English corrections.

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

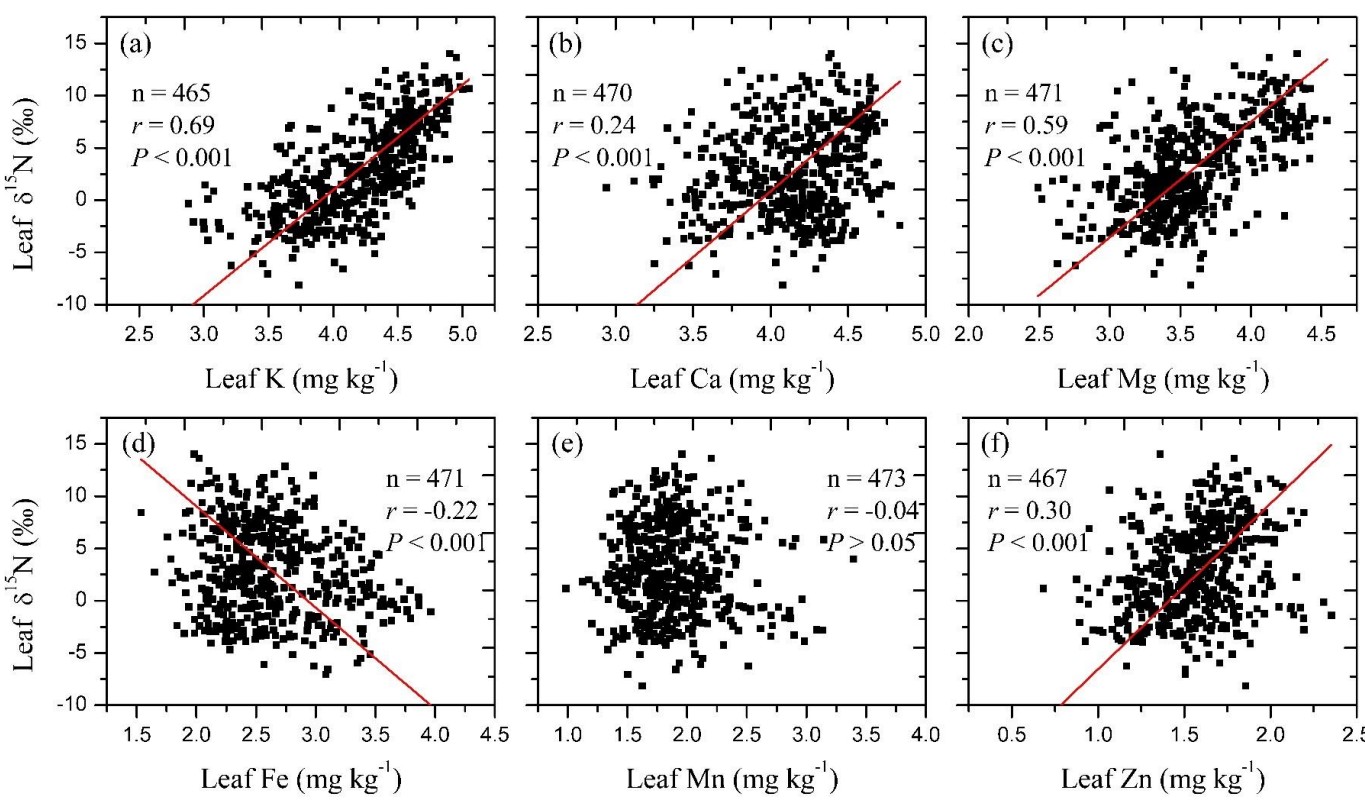

Fig. 1 Relationships between leaf $\delta^{15}N$ and the contents of leaf K (a), Ca (b), Mg (c), Fe (d), Mn (e) and Zn (f) for all non-$N_2$-fixing species pooled together. The contents of leaf metallic elements were log-transformed.


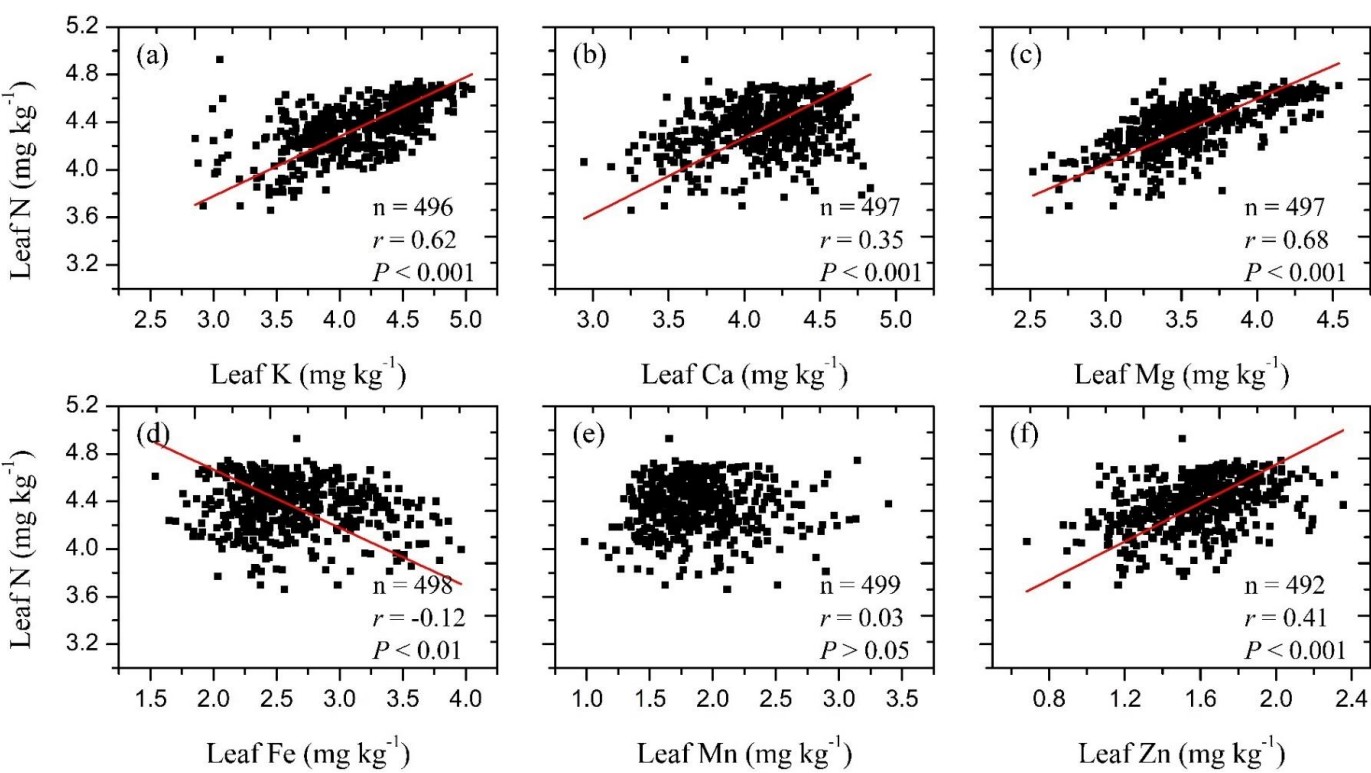

Fig. 2 Relationships between the content of leaf N and the contents of leaf K (a), Ca (b), Mg (c), Fe (d), Mn (e) and Zn (f)

for all non-N$_2$-fixing species pooled together. The contents of leaf metallic elements were log-transformed.





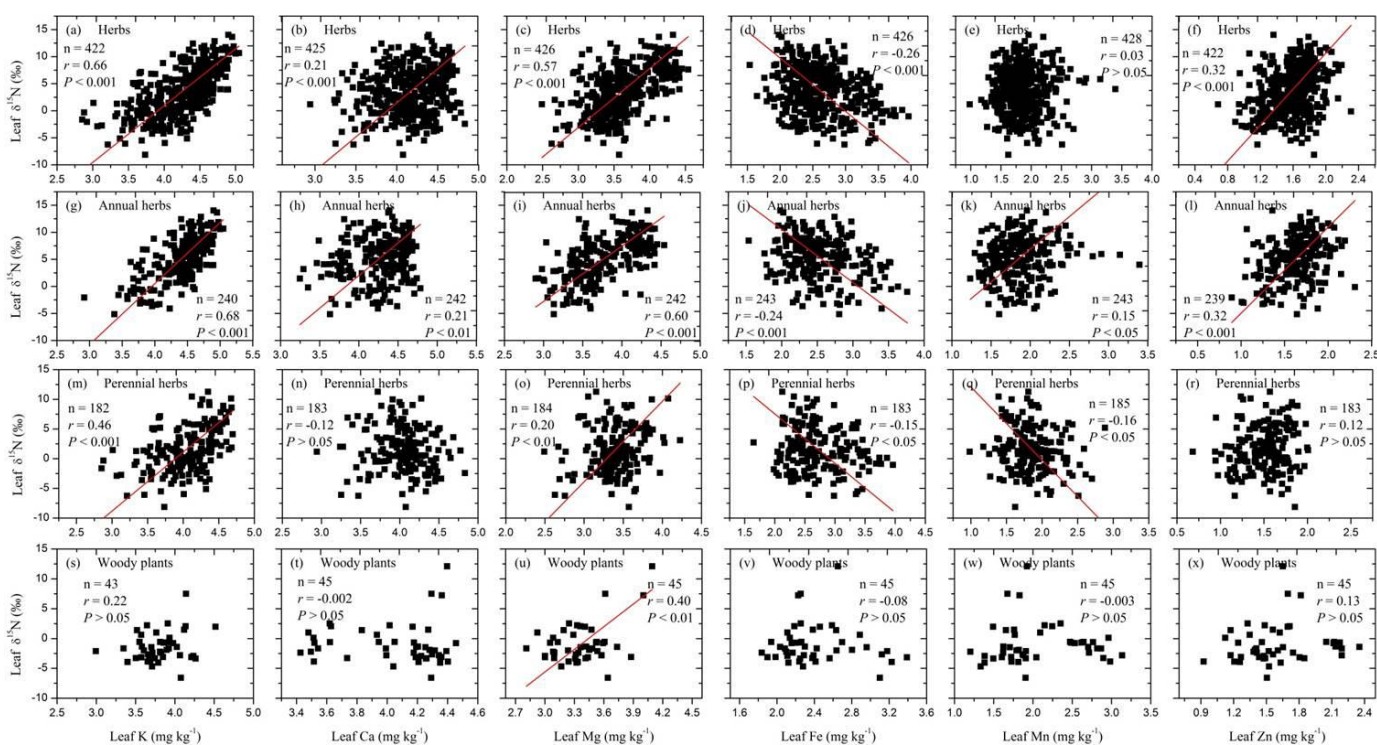

Fig. 3 Relationships between leaf $\delta^{15}$N and the contents of leaf metallic elements for herbs (a-f), annual herbs (g-l), perennial herbs (m-r) and woody plants (s-x). The contents of leaf metallic elements were log-transformed.


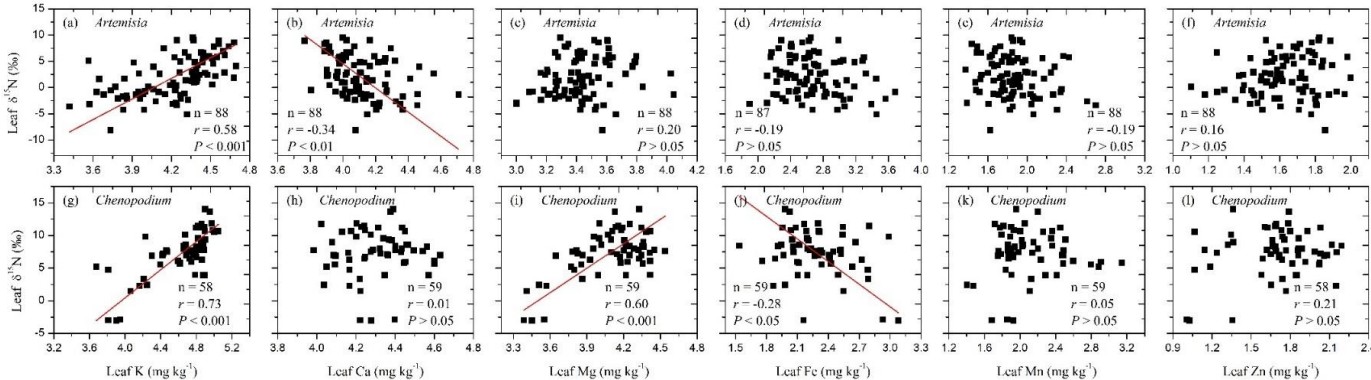

Fig. 4 Relationships between leaf δ$^{15}$N and the contents of leaf metallic elements for two common genera (*Artemisia* (a-f)

and *Chenopodium* (g-l)) across the sampling region. The contents of leaf metallic elements were log-transformed.

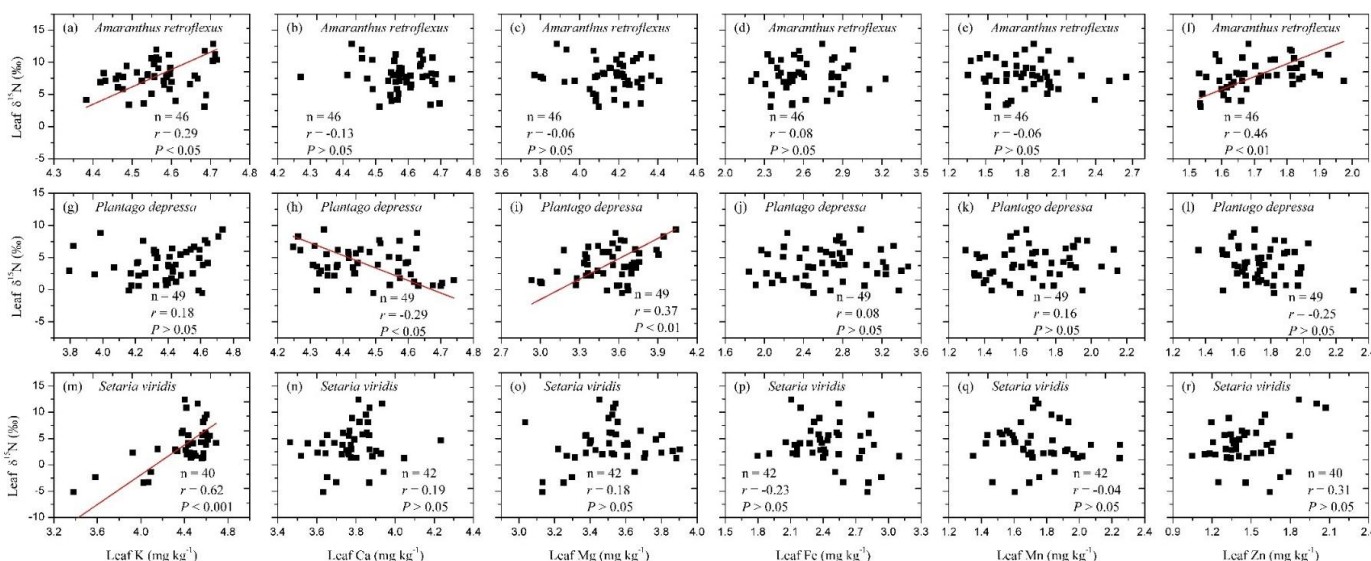

Fig. 5 Relationships between leaf δ¹⁵N and the contents of leaf metallic elements for three widespread plant species (*Amaranthus retroflexus* (a-f), *Plantago depressa* (g-l) and *Setaria viridis* (m-r)) across the sampling region. The contents of leaf metallic elements were log-transformed.



Table 1. Multiple linear regressions of leaf $\delta^{15}N$ against leaf metallic nutrients based on ordinary least-square (OLS) estimation.

| Model | $R^2$ | Adjust $R^2$ | $P$ |
|-------|-------|--------------|------|
| 1 | 0.543 | 0.540 | <0.001 |
| 2 | 0.150 | 0.145 | <0.001 |
| 3 | 0.557 | 0.551 | <0.001 |

*Note*: Model-1 is the multiple regression of leaf $\delta^{15}N$ against leaf K, Ca and Mg; Model-2 is the multiple regression of leaf $\delta^{15}N$ against leaf Fe, Mn and Zn. Model-3 is the multiple regression of leaf $\delta^{15}N$ against leaf K, Ca, Mg, Fe, Mn and Zn.



506

Table 2. Comparison of the results of bivariate correlation analysis with partial correlation analyses of leaf $\delta^{15}N$ vs. leaf metallic nutrients after controlling for vegetation type and/or soil type.

509

| Controlled factors | K | | Ca | | Mg | | Fe | | Mn | | Zn | |
|---|---|---|---|---|---|---|---|---|---|---|---|---|
| | $r$ | $P$ | $r$ | $P$ | $r$ | $P$ | $r$ | $P$ | $r$ | $P$ | $r$ | $P$ |
| None | 0.689 | <0.001 | 0.243 | <0.001 | 0.587 | <0.001 | -0.223 | <0.001 | -0.036 | >0.05 | 0.296 | <0.001 |
| Vegetation type | 0.690 | <0.001 | 0.235 | <0.001 | 0.588 | <0.001 | -0.223 | <0.001 | -0.035 | >0.05 | 0.293 | <0.001 |
| Soil type | 0.675 | <0.001 | 0.240 | <0.001 | 0.589 | <0.001 | -0.196 | <0.001 | -0.095 | <0.05 | 0.251 | <0.001 |
| Vegetation & Soil type | 0.674 | <0.001 | 0.239 | <0.001 | 0.590 | <0.001 | -0.195 | <0.001 | -0.101 | <0.05 | 0.251 | <0.001 |

510