# Peer review of "Relationships between leaf $\delta^{15}N$ and leaf metallic nutrients 1 Chongjuan Chen1,2 Yingjie Wu1 Shuhan Wang3 Zhaotong Liu1 Guoan Wang1\* 2 1Beijing Key Laboratory of Farmland Soil Pollution Prevention and Remediation, Departm"

_Biogeosciences, 2019_

## Referee Comment (RC1) · Anonymous Referee #1 · 27 Dec 2019

The manuscript reported large amount of valuable data. The result and discussion are generally reliable and reasonable. However, please check all the data carefully. For example, in Table S5, r value for $\delta$15N vs pH is in bold font but its p value is larger than 0.05.
* * *

---

## Referee Comment (RC2) · Anonymous Referee #2 · 30 Dec 2019

Chen and coauthors investigated the relationships between leaf $\delta$15N and metallic nutrients across a large number of sites from northeast to southwest China. They found leaf $\delta$15N was positively correlated with leaf K, Ca, Mg and Zn but negatively correlated with leaf Fe, and these correlations were not affected by vegetation and soil type. However, the relationship between $\delta$15N and Mn was dependent on vegetation and soil type. This is an interesting study which examined the relationships between leaf $\delta$15N and leaf metallic nutrients for the first time. I have no major concerns with the content of the manuscript. However, the paper needs a strong hand in English editing. The language is often not precise/exact, sometimes appears to be ordinary and has grammatical errors. I strongly suggest to critically check the grammars, read and polish the manuscript using exact language. I listed some examples below but more need to

[Figure]

be revised. I would recommend for publishing if this and the following comments could be addressed.

Specific comments: Line 15: "Calcium" should be "calcium". Line 25: meteallic → metallic Line 29: Should be attention. Attention is an uncountable noun. Line 29-32: Need to be rephased. Delete "deemed as". This sentence is too long. Split to two sentences. Line 34: Change to "Revealing the potential factors that influence leaf $\delta$15N and investigating the relationships between them could help improve our current understanding of N cycling". Line 37: Should be "Much attention has". Line 43: Change "have exposed" to "demonstrated". Line 46-47: To our knowledge, no report exists for the relationships... Line 55: "plays a vital" Line 52-66: This paragraph is poorly written and needs to be revised. Line 80-88: Where is the description for soil sampling? Line 164: should be "positively correlated to leaf K..." Line 161-175: Check the use of "while" and "whereas". Both of them are conjunction and should connect two sentences. Line 224: "Whereas"→ "Nevertheless" Line 255: "but also served as ..." Line 261: "leaf and roots, and then..." Line 277-278: Are there any correlations among metallic nutrients? The different relationships observed for Zn, Mn, Fe are very interesting and deserve further discussion. Line 290-292: ...leaf metallic nutrients almost did not change..., which suggested ...

---

## Author Comment (AC1) · 16 Jan 2020

Comment: The manuscript reported large amount of valuable data. The result and discussion are generally reliable and reasonable. However, please check all the data carefully. For example, in Table S5, r value for $\delta$15N vs pH is in bold font but its p value is larger than 0.05. Answer: Thanks very much for your evaluation. We are very sorry for this mistake. We have corrected it (See Table S5) and thoroughly checked the data.

Please also note the supplement to this comment:
https://www.biogeosciences-discuss.net/bg-2019-328/bg-2019-328-AC1-supplement.pdf

[Figure]

**Supplement:**

**Answers to the questions:**
* * *
**Reviewer #1:**

**1. Comment:** The manuscript reported large amount of valuable data. The result and discussion are generally reliable and reasonable. However, please check all the data carefully. For example, in Table S5, $r$ value for $\delta^{15}N$ vs pH is in bold font but its $p$ value is larger than 0.05.

**Answer:** Thanks very much for your evaluation. We are very sorry for this mistake. We have corrected it (See Table S5) and thoroughly checked the data.

Table S5. The correlations between leaf $\delta^{15}N$ and soil indexes.

| Correlations | $r$ | $P$ |
|---|---|---|
| Leaf $\delta^{15}N$ vs. soil $\delta^{15}N$ | **0.506** | < 0.01 |
| Leaf $\delta^{15}N$ vs. SOM | -0.010 | > 0.05 |
| Leaf $\delta^{15}N$ vs. pH | 0.031 | > 0.05 |
| Leaf $\delta^{15}N$ vs. soil C/N | 0.120 | < 0.05 |
| Leaf $\delta^{15}N$ vs. soil N | -0.074 | > 0.05 |
| Leaf $\delta^{15}N$ vs. soil density | 0.102 | > 0.05 |
| Leaf $\delta^{15}N$ vs. soil sand | 0.007 | > 0.05 |
| Leaf $\delta^{15}N$ vs. soil silt | -0.010 | > 0.05 |
| Leaf $\delta^{15}N$ vs. soil clay | 0.033 | > 0.05 |
| Leaf $\delta^{15}N$ vs. soil silt/clay | 0.092 | > 0.05 |

---

## Author Comment (AC2) · 16 Jan 2020

1. Comment: Chen and coauthors investigated the relationships between leaf $\delta$15N and metallic nutrients across a large number of sites from northeast to southwest China. They found leaf $\delta$15N was positively correlated with leaf K, Ca, Mg and Zn but negatively correlated with leaf Fe, and these correlations were not affected by vegetation and soil type. However, the relationship between $\delta$15N and Mn was dependent on vegetation and soil type. This is an interesting study which examined the relationships between leaf $\delta$15N and leaf metallic nutrients for the first time. I have no major concerns with the content of the manuscript. However, the paper needs a strong hand in English editing. The language is often not precise/exact, sometimes appears to be ordinary and has grammatical errors. I strongly suggest to critically check the grammars, read and polish the manuscript using exact language. I listed some examples below but more need to be revised. I would recommend for publishing if this and the following comments could be addressed. Answer: Thanks for your comments. We have corrected the grammatical errors according to your comments, and we will send our manuscript to professional company for polishing.

2. Specific comments: (1) Line 15: "Calcium" should be "calcium". Answer: Corrected. (2) Line 25: meteallic →metallic. Answer: Corrected. (3) Line 29: Should be attention. Attention is an uncountable noun. Answer: Very thanks. Corrected. (4) Line 29-32: Need to be rephased. Delete "deemed as". This sentence is too long. Split to two sentences. Answer: Very thanks. Corrected as follows, Nitrogen (N) cycling has received considerable attention, because N is the key element in regulating productivity of terrestrial ecosystems (Fay et al., 2015; Wieder et al., 2015) and many nitrogenous compounds generating from N cycling are associated with major environment issues (Bourgeois et al., 2018; Desmit et al., 2018). (5) Line 34: Change to "Revealing the potential factors that influence leaf $\delta$15N and investigating the relationships between them could help improve our current understanding of N cycling". Answer: Corrected. (6) Line 37: Should be "Much attention has". Answer: Corrected. (7) Line 43: Change "have exposed" to "demonstrated". Answer: Corrected. (8) Line 46-47: To our knowledge, no report exists for the relationships... Answer: Corrected. (9) Line 55: "plays a vital". Answer: Corrected. (10) Line 52-66: This paragraph is poorly written and needs to be revised. Answer: Very thanks. Corrected as follows, K is the activator of many enzymes, it promotes photosynthesis and has important influences on nitrate and ammonium utilization in plants (Coskun et al., 2017; Zhang et al., 2010). Ca is involved in and plays a vital role in nitrate signaling network (Krouk et al., 2017; Liu et al., 2017). Mg is an essential component of chlorophyll and associated closely with nitrate reduction process in plants (Bose et al., 2011). Fe participates in many physiological processes in plants, such as nitrogen assimilation, photosynthesis, respiration, DNA synthesis, dinitrogenase synthesis (Balk and Pilon, 2011; Shokrollahi et al., 2018). Mn and Zn are the important components of enzymes, which are included

in carbohydrate metabolism, N metabolism and RNA synthetase (Mukhopadhyay and Sharma, 1991; Henriques et al., 2012). Accordingly, these metallic elements are essential for plant N metabolism processes. Because N metabolism processes, including acquire different N forms, nitrate and ammonium assimilation and allocation, could cause isotopic fractionation (Evans, 2001; Tcherkez and Hodges, 2008; Liu et al., 2014), we hypothesized that leaf $\delta$15N should be related to the contents of these metallic nutrients. Thus, in order to confirm this hypothesis, we sampled more than 600 plant samples from mainland China and analyzed leaf $\delta$15N and contents of leaf K, Ca, Mg, Fe, Mn and Zn. (11) Line 80-88: Where is the description for soil sampling? Answer: We are very sorry for that. We have changed "Plant and soil sampling" into "Plant sampling", and considering that plant metallic nutrients and $\delta$15N should be the main content, we have added the description for soil sampling and measurements in the Supplementary Information. The description of soil sampling and measurements were as follows, Surface mineral soils (0 – 5 cm) were sampled after removing the litter layer. At each location, three squares (0.5 m $\times$ 0.5 m) within a 200 m2 area were set to collect the mineral soils. Samples were sieved through a 2 mm sieve to remove stones and plant residues in soil. Soil samples were air-dried and divided into three parts. First part of samples was used to determine soil $\delta$15N and N content. Similar to the measurement of $\delta$15N and N content in leaves, the dried soil sample was ground into fine powder, and ïAď15N and N contents of soils were also determined by a DeltaPlus XP mass spectrometer coupled with an automated elemental analyzer in a continuous flow mode. The second part of soil sample was used to determined soil texture (clay, silt and sand content) and pH. Soil texture was analyzed using a particle size analyzer (Malvern Masterizer 2000, UK) after removing the calcium carbonates and organic matter. Soil pH was analyzed by a pH electrode in soil-water suspension, with a soil/water ratio was 0.4 (10 g soil and 25 mL deionized water without carbon dioxide). The third part of soil sample was treated to determine the soil organic carbon (SOC) content, soil sample was immersed in 1 mol/L HCl for 24 h and was stirred four times to remove carbonate, then, the soil sample was washed to neutrality

using distilled water and oven-dried at 50 °C. In addition, at each location, another three soil samples were collected using a ring to measure soil bulk density. Soil bulk density was determined after oven-drying at $105 \pm 2$ °C to a constant weight, and it was the dry weight divided by the certain volume of the ring knife. (12) Line 164: should be "positively correlated to leaf K...". Answer: Corrected. (13) Line 161-175: Check the use of "while" and "whereas". Both of them are conjunction and should connect two sentences. Answer: Thanks very much. We have checked them. (14) Line 224: "Whereas"→ "Nevertheless". Answer: Corrected. (15) Line 255: "but also served as...". Answer: Corrected. (16) Line 261: "leaf and roots, and then...". Answer: Corrected. (17) Line 277-278: Are there any correlations among metallic nutrients? The different relationships observed for Zn, Mn, Fe are very interesting and deserve further discussion. Answer: Thanks. As showed in Table S2, Almost all elements are related to each other. The relationships between leaf $\delta 15N$ and leaf Fe, Mn, and Zn did need further discussion. According to your comments, we add some discussions about this part (as follows), but it is still not enough and need further exploration. Fe, Mn and Zn participate in N assimilation processes, even Fe and Zn are the key components of enzymes in nitrogen metabolism (Fischer et al., 2005; Henriques et al., 2012; Ventura et al., 2013). The reason why the relationship between $\delta 15N$ and Fe is opposite to the relationship between $\delta 15N$ and Zn might be related to the different roles of the two elements in plant N assimilation. Iron-sulfur protein is the electron donor for nitrate reductase, and Fe supply has important influences on nitrate reductase activity and then on $NO3-$ utilization (Pandey, 2000). Zn is the component of glutamic dehydrogenase and might mediate the $NH4+$ utilization (Kitagishi and Obata, 1986). We do not know the underlying mechanisms as yet. Mn can promote the synthesis of chlorophyll and assimilation of nitrate (Dučič and Polle, 2005), we cannot also explain the non-significant relationship between leaf Mn and $\delta 15N$. Line 290-292: ...leaf metallic nutrients almost did not change..., which suggested ... Answer: Thanks for your detailed comments. We have corrected them.

Please also note the supplement to this comment:
https://www.biogeosciences-discuss.net/bg-2019-328/bg-2019-328-AC2-
supplement.pdf

———————————————————

---

## Referee Comment (RC3) · Anonymous Referee #3 · 25 Feb 2020

Chen et al. used a long transect dataset to reveal the relationships between leaf delta 15N and leaf metallic nutrients. This study is good and dataset is valuable. I have few comments. 1) It should give a more general introduction about the leaf 15N. What is the meaning of leaf 15N value can be given? 2) Why did not include the climate factors into the analysis? As we know the climatic factors are the most important factors in controlling leaf isotope values. 3) The AIC value is more useful to select which modeling is more powerful to build the relationships between isotope values and nutrients (Tables 1)

---

## Author Comment (AC3) · 28 Feb 2020

**Answers to the questions:**
* * *
**Reviewer #3:**

**1. Comment:** Chen et al. used a long transect dataset to reveal the relationships between leaf delta15N and leaf metallic nutrients. This study is good and dataset is valuable.

**Answer:** Thanks for your comments.

**2. Comment:** It should give a more general introduction about the leaf 15N. What is the meaning of leaf 15N value can be given?

**Answer:** Thanks for your comments. We have added the statement about leaf $\delta^{15}N$(mark in blue color)in the introduction section as follows,

Nitrogen (N) cycling has received considerable attention, because N is the key element in regulating productivity of terrestrial ecosystems (Fay et al., 2015; Wieder et al., 2015) and many nitrogenous compounds generating from N cycling are associated with major environment issues (Bourgeois et al., 2018; Desmit et al., 2018). Natural N isotopic technique has been widely taken as a powerful tool in exploring the N biogeochemical cycling (Evans, 2001; Robinson, 2001), and nitrogen isotopic composition ($\delta^{15}N$) in leaf has been usually regarded as an integrator of N cycling (Evans, 2001; Robinson, 2001; Houlton et al., 2006, 2007; McLauchlan et al., 2007, 2013). This might be associated with the fact that leaf $\delta^{15}N$ could indicate soil N availability (Craine et al., 2009; Högberg et al., 2011) and plant N utilization strategies (Kolb and Evans, 2002; Houlton et al., 2007). Thus, revealing the potential factors that influence leaf $\delta^{15}N$ and investigating the relationships between these factors and leaf $\delta^{15}N$ could help improve our current understanding of N cycling (Craine et al., 2009; Hobbie and Högberg, 2012).

**3. Comment:** Why did not include the climate factors into the analysis? As we know the climatic factors are the most important factors in controlling leaf isotope values.

**Answer:** Yes, as we had mentioned in the introduction section, climatic factors, such

as temperature and precipitation, have been reported to exert important influences on the variations in leaf $\delta^{15}$N. However, we have emphatically explored the relationships between leaf $\delta^{15}$N and temperature and precipitation in another manuscript (Chen et al., under review). So in this manuscript, we just focused on the relationships between leaf $\delta^{15}$N and metallic nutrients and did not include the climate factors.

**4. Comment:** The AIC value is more useful to select which modeling is more powerful to build the relationships between isotope values and nutrients (Tables 1).

**Answer:** Thanks for your comments. We have calculated the *AIC* values of each model as in Table 1. The model 3 has the lowest *AIC* value of 1058.3, thus, the model 3 is the best for explaining the variations of leaf $\delta^{15}$N. Meanwhile, we find that the *AIC* value of model 3 is only slightly higher than that of model 1, which means adding leaf Fe, Mn and Zn could not significantly improve the goodness of fit of the model 3 compared with that of model 1. Thus, the results of *AIC* values are also consistent with the results of $R^2$ or adjust $R^2$, i.e. the $R^2$ or adjust $R^2$ of model 3 is only a little larger than that of model 1. Comparison of the two models shows that leaf K, Ca, Mg are the major factors the main driving factors. of leaf $\delta^{15}$N.

Table 1. Multiple linear regressions of leaf $\delta^{15}$N against leaf metallic nutrients based on ordinary least-square (OLS) estimation.

| Model | $R^2$ | Adjust $R^2$ | *AIC* | *P* |
|---|---|---|---|---|
| 1 | 0.543 | 0.540 | 1091.5 | <0.001 |
| 2 | 0.150 | 0.145 | 1404.6 | <0.001 |
| 3 | 0.557 | 0.551 | 1058.3 | <0.001 |

*Note*: Model-1 is the multiple regression of leaf $\delta^{15}$N against leaf K, Ca and Mg; Model-2 is the multiple regression of leaf $\delta^{15}$N against leaf Fe, Mn and Zn. Model-3 is the multiple regression of leaf $\delta^{15}$N against leaf K, Ca, Mg, Fe, Mn and Zn. AIC, Akaike Information Criterion.